# Linking Supply Chain Disruption Orientation to Supply Chain Resilience and Market Performance with the Stimulus–Organism–Response Model

Aaron Rae Stephens [1], Minhyo Kang [2] and Charles Arthur Robb [3,*]

1 Department of Business Administration and Accounting, Hartwick College, Oneonta, NY 13820, USA; stephensa@hartwick.edu
2 Department of Asian Studies, Busan University of Foreign Studies, Busan 46234, Korea; minhyo74@bufs.ac.kr
3 Department of Business Administration, Dongguk University, WISE Campus, Gyeongju 38066, Korea
* Correspondence: charles@dongguk.ac.kr

**Abstract:** Since 2020, supply chain disruptions have emerged as an ever-present challenge. This research provides a glimpse into the organizational structures that develop supply chain resilience and market performance amid continuous supply chain disruptions. Utilizing psychosomatic variables and empirical modeling, a model was constructed through a review of extant literature and tested with PLS-SEM analysis. Uniquely, this research model is framed with the stimulus–organism–response model; thus, placing a firm within the context of a tumultuous environment where stimuli elicit responses from an organization that behaves as an organism. Results demonstrate that organizational culture plays a critical role in developing supply chain resilience amid supply chain dynamism. Market performance was also developed but only through supply chain resilience; supply chain disruption orientation alone did not improve market performance. Mediation effects highlight the importance of supply chain disruption orientation, a strategic orientation that cements an organization's ability to develop supply chain resilience.

**Keywords:** supply chain dynamism; supply chain disruption orientation; supply chain resilience; market performance; stimulus–organism–response model

## 1. Introduction

Cost pressures experienced by firms in international markets through increased environmental dynamism have led to a further reliance by organizations on outsourcing and offshoring strategies within many manufacturing and R&D activities (Yu et al. 2019); greatly adding to the complexity of global supply chains (Katsaliaki et al. 2021). Supplementary to this, changes in the global commercial environment since the onset of the COVID-19 pandemic have introduced major challenges to businesses around the world, including issues related to halted factory production and disrupted supply chains (Laato et al. 2020). The pandemic has also exposed novel risks in supply chains that will require creative perseverance over the short and long-term. Rai (2020) notes that the pandemic has established a mandate for organizations to rapidly rethink their value-creation models for an immensely different global context (Katsaliaki et al. 2021).

For global supply chains, additional pressures have been endured due to exaggerated imbalances between supply and demand factors, as well as panic buying trends in consumer consumption (Addo et al. 2020; Wong et al. 2020). While organizations assess methods that best maintain or build cost competitiveness by way of managing their supply chains (Katsaliaki et al. 2021), disruptions remain an inevitable by-product of extended supply lines influenced by negative, unanticipated events such as the ones experienced under the current environmental situation (Al-Hakimi et al. 2021). Moreover, disruptions to a supply chain are noted as causing performance issues for firms (Golgeci and Ponomarov 2014).

Therefore, the preservation and strengthening of supply chains remains an important factor for consideration that has both practical and theoretical implications (Wong et al. 2020). While the idea of conditioning a supply chain seems a palpable consideration to undertake, this strategy may be dissimilar between organizations due to issues such as limited resources and inadequate firm capabilities (Polyviou et al. 2020). While there is literature that exists regarding supply chain dynamism and reliance, the introduction of established theories to support the relationship of the above-mentioned constructs with firm performance is encouraged (Katsaliaki et al. 2021). Furthermore, Polyviou et al. (2020) concluded that empirical theory supporting the notion of resilience in varying supply chains studies should require additional insight (Shashi et al. 2020).

To establish a more holistic and novel approach to the literature regarding supply chain theory and for addressing this research disparity, the current study aims to center the literature review and study results around three key components. First, the authors utilize SEM analysis and a framework built around a psychological tool to explain supply chain disruptions in the context of organizational actors. Second, the research responds to the request by authors (Katsaliaki et al. 2021) to explore supply chain reliance in greater detail. Finally, the current research investigates both the direct and mediating effects of supply chain disruption orientation and supply chain resilience, providing practical and theoretical implications for supply chain literature. Scientific implications include a firm-level survey, a new approach to framing supply chain empirical research, and evidence of organizational relationships within supply chain management. Practitioner implications focus on improving organizational culture in order to develop supply chain resilience and market performance.

To design the research framework, the stimulus–organism–response (S-O-R) framework (Mehrabian and Russell 1974) is expanded upon. While the SOR model has not been employed before in supply chain research, it presents a unique approach to investigate how the stimuli of dynamism in supply chains during COVID-19 leads to an organism of disruption orientation in organizations and ultimately directs these organisms in the environment to enlist a response in firms toward practicing supply chain resilience (Katsaliaki et al. 2021; Matos and Krielow 2019; Robb and Stephens 2021).

As the COVID-19 pandemic situation is considered an exceptional event, research related to the pandemic permits greater insight into firm behavior during a global pandemic event characterized by uncertainty. Shashi et al. (2020) noted that research studies regarding the effects of the pandemic on the marketplace are lacking, making it more difficult for managers to fully comprehend the impact of supply chain disturbance. Moreover, supply chain disruptions and resilience have of late become research areas demonstrating much academic interest as firms contend with a lack of information and knowledge sharing during the pandemic (Katsaliaki et al. 2021; Shashi et al. 2020). To comprehend the above literature, the current research tests a model with data collected from organizations in the United States during the month of April 2021. The United States presents an intriguing sample base to study as it remains a nation heavily influenced by supply chain resilience. Rai (2020) remarked that the country is dealing with a 14.7% unemployment rate since the pandemic began, which is its highest job loss rate since the Great Depression. Interestingly, while the unemployment rate has increased in the U.S., the employment rate has continued to decrease as numerous individuals choose not to return to work (or take early retirement); this has amplified disruptions to supply chain operations throughout the U.S. (Katsaliaki et al. 2021; Rai 2020). As individuals critical to the success of global supply chains (e.g., truckers, logistics and distribution personnel, and inventory specialists, etc.) indicate their intentions not to return to the workforce, the pandemic has unambiguously exposed the need to build resilience in ways that both save lives and preserve livelihoods (Rai 2020). Thus, the context of the study provides an avenue in which to explore the remaining two components of the study and consider the extent to which the research constructs contribute toward market performance.

To further explore the motions presented above, the remainder of the paper is structured accordingly. A review of current literature focused on the SOR model and supply chain constructs will follow. Thereafter, theories are presented to test hypotheses and establish an empirical research model. Subsequently, data collection methods and results are reported. To conclude the paper, practical and theoretical implications, limitations, and future work are offered.

## 2. Literature Review

### 2.1. Theoretical Underpinning

While contemporary research in the field of supply chain management provides clear theoretical connections between theory and practice (Polyviou et al. 2020), particular attention has been given to facilitate the void existing between supply chain research and the components of an organization's external environment (Shashi et al. 2020). Given the highly dynamic nature of supply chains today, it is necessary to adopt a theoretical framework that fits with the organic life-like behavior that has evolved out of such dynamism. The SOR framework seems to be an ideal theory as it frames the organization as an organism within a dynamic, constantly changing environment, comparable with firms operating within tumultuous supply chains.

Initially proposed by Mehrabian and Russell (1974) as a framework to conceptualize consumer or firm behavior in various situations, the SOR model has remained a theory utilized by researchers to better understand factors related to particular organizational settings. While the SOR model has primarily contributed to research as a psychological tool developed to measure individual behavior (Mehrabian and Russell 1974; Matos and Krielow 2019), novel research on the model has reinforced the explanative competencies of the model when used to measure aspects of organizational performance (Li et al. 2020) and the propinquity between the environment and the firm (Matos and Krielow 2019).

In the current research study, exploiting the SOR model tolerates for a far greater reflection of supply chain research in three particular areas. First, the stimulus component of the model (or extrinsic factors) influences the behavior and decision-making aspects of a firm; second, the organism or internal processes of a firm refers to an orientation in the organization that will eventually lead to a response based on the stimulus; finally, the response component of the model denotes an intention by the firm to reciprocate toward 'changes' in the background (Robb and Stephens 2021).

According to Robb and Stephens (2021), stimulus is concerned with behavior occurring in the competitive domain that influences the internal state of the organization. Due to the nature of this behavior surrounding the stimulus, organizations would invariably progress toward remodeling internal processes (both cognitive and affective) to assist the organization in attaining improved value in its reaction to environmental stimuli. Under this scenario, supply chain disruption orientation acts as a firm organism, orientating an organization's operations and processes toward becoming concerned with successfully responding to situational factors (Li et al. 2020). Thus, an organism (within the SOR framework) acts as a platform or connection with which an organization can provide a level of justification for the way in which it responds to its competitive context and makes decisions that would have an impact on the firm's performance (Matos and Krielow 2019). Therefore, supply chain resilience has been linked to the 'response' aspect of the SOR model. Research indicates that organizations eager to recover operations following a disruption in the supply chain can rapidly recover when organizational culture is properly positioned (Ambulkar et al. 2015; Chowdhury and Quaddus 2017). Finally, to consolidate the research model, market-related performance is regarded as an important measurement of firm success in the current research. Consequently, the inclusion of a performance aspect to the SOR model provides supplementary interpretations of the study results (Gotteland et al. 2020) and a natural progression in the supply chain literature (Li et al. 2020).

## 2.2. Supply Chain Dynamism

Recent dramatic events such as COVID-19 have caused major behavioral changes in the environments within which global supply chains exist. For example, according to event systems theory (Morgeson et al. 2015), intrusive events (such as COVID-19) have been noted as causing multifaceted inconsistencies, which have become 'far too' common in the practices of organizations (Reimann et al. 2017). Currently, contingencies and firm dilemmas brought about by the pandemic have exposed weaknesses related to lags in information and the fragility associated to supply chains worldwide (Rai 2020). This adjustment in processes and products in organizations as a derivative of the environment is referred to as supply chain dynamism. Zhou and Benton (2007) denoted that dynamism in supply chains is characterized by changes in the pace of either organizational operating processes or the degree of innovation frequency for services and products.

Craighead et al. (2020) as an example found that change in the supply chain of firms led to irregular outcomes for these organizations. Consequently, research regarding dynamism in supply chain studies has garnered exceptional attention recently (Wong et al. 2020), as knowledge acquisition regarding supply chain dynamism has been found to contribute and encourage the efficient adoption of supply chain activities in organizations (Lee et al. 2016). For organizations engaged in the struggles of the current business cycle, unforeseen interruptions within the supply chain have become regular occurrences, resulting in economic losses or company insolvency (Rai 2020; Scholten et al. 2014). Therefore, the exploration of supply chain dynamism remains an integral addition to supply chain literature, which contributes to knowledge related to competitiveness (Scholten et al. 2014) and the subsequent activities implemented by organizations to achieve performance benefits (Lee et al. 2016; Wong et al. 2020).

## 2.3. Supply Chain Disruption Orientation

Since the onset of globalization, supply chains around the world have repeatedly been challenged by the ever-increasing and complex nature of relationships throughout these channels. The inception of the COVID-19 pandemic has propagated this issue further, as organizations around the globe have been forced to temporarily halt or taper off their operations (Craighead et al. 2020). Throughout this period of uncertainty, a growing number of organizations have begun to familiarize themselves with various operations and processes with which they are able to greatly manage disruptions in their supply chains (Bode et al. 2011). Disruptions are categorized as uncertain events that interrupt the regular flow of goods and services within the supply chain (Craighead et al. 2020). The ability of becoming aware of impending disruptions has been defined as supply chain disruption orientation (SCDO), where organizations have indirectly created scenario-style planning during these disruptions to learn and scrutinize their approaches to these unforeseen events (Ambulkar et al. 2015). Approaches linked to a strong SCDO include the reintroduction or straightening of risk management infrastructure (Bode et al. 2011) or learning from previous disruptions as a method of lessening future threats (Reimann et al. 2017) and exploiting new opportunities (Ambulkar et al. 2015). By utilizing a better understanding of dynamism in an environment and a progression in the direction of orientating an organization to consider strategies geared at mitigating changes in the environment, the current research considers the following.

**H1.** *Firms that experience a high degree of supply chain dynamism also develop supply chain disruption orientation as a response.*

While research on disruption orientation seems to confirm that the process of managing disruptions in the supply chain could benefit organizational operations (Ambulkar et al. 2015), less consistency exists with regards to conformity in the relationship between SCDO and market performance (Chopra and Meindl 2004). While a strategic orientation geared toward understanding the future direction of business operations may constitute progressive thinking on the part of organizations, these strategic steps may not necessarily lead to

improved firm performance. For example, Yu et al. (2019) found that a focused mindset on performance in the context of supply chain operations could have unintended consequences on supply chain partners, which would then reciprocate negative outcomes for the origin firm. Bode et al. (2011) also emphasized that a vigorous supply chain disruption orientation led to a stronger motivation to act in the wake of a disturbance. Yu et al. (2019) suggest that organizations should rather assume an external focus when measuring market performance and consider the impact of organizational strategies on other supply chain partners. This view is supported by Chopra and Meindl (2004) who denoted that the performance of an organizations' supply chain would benefit greatly by considering the indirect impact a firm would have on the overall supply chain (Wong et al. 2020). With consideration to the above statement, this research contemplates the next hypothesis:

**H2.** *Supply chain disruption orientation has a positive and significant impact on market performance.*

As firms identify the most efficient direction in which to proceed during, or following a major disruption, the issue of resilience often surfaces (Ambulkar et al. 2015). Queiroz et al. (2021) stressed that supply chain disruption orientation could help firms develop supply chain resilience after a thorough investigation of 112 Brazilian companies. Resilience in a supply chain follows a concerted effort by an organization to manage any, and all disruptions in a supply chain. Resilience is concerned with the development of capacities to recover from, and mitigate disruptions in advance (Chowdhury and Quaddus 2017). Blackhurst et al. (2005) mention the importance of supply chain resilience, as it is concerned with the ability of firms to improve their supply chain operations from unanticipated disruptions. Under this logic, organizations would utilize their SCDO to forecast, mediate, and formulate supply chain operations to build upon their abilities to manage the aftereffects of a disruption. Subsequently, the subsequent hypothesis is presented as follows.

**H3.** *Organizational cultures that reflect supply chain disruption orientation will positively and significantly lead to supply chain resilience.*

*2.4. Supply Chain Resilience*

Incongruities in the business environment require firms to better allocate operating resources to manage supply chain issues. This assignment of resources to mitigate irregularities is referred to as the organizations supply chain resilience (Annarelli and Nonino 2015). This competence enables a supply chain to adapt and quickly respond to events that are random in nature (Ambulkar et al. 2015). Similarly, Ponomarov and Holcomb (2009) defined resilience in a supply chain as a capability related to maintain attentiveness for unexpected events (Zailani et al. 2015). Moreover, supply chain resilience incorporates an ability to respond and recover from disruptions while maintaining efficient operations in an organization. Chowdhury and Quaddus (2017) noted that the competency of a firm to reduce the impact of a disruption and recover organizational processes to their original levels (and possibly improve firm performance) renders supply chain resilience a dynamic capability (Yu et al. 2019). Thus, the ability of an organization to adapt, response, and recover from internal and external disruptions can increase a firm's competitive advantage and overall performance (Yu et al. 2019). Due to the nature of supply chain resilience as being a capability associated with the sustainability and longevity of the supply chain (Ponomarov and Holcomb 2009), prior studies assumed a positive impact between supply chain resilience and firm performance outcomes (Wong et al. 2020). Therefore, the current research presents one final hypothesis to incorporate a performance aspect into the SOR model and conceptualization of the study framework as follows.

**H4.** *Supply chain resilience has a positive and significant impact on market performance as firms are better able to serve their customers.*

### 3. Methodology

This empirical study utilizes a model framed by extant literature that is tested by using structural equation modelling (SEM); moreover, psychometric constructs are represented by questions that in sum create the variables and the model. Several methods for SEM analysis are available; however, PLS-SEM is the most appropriate method for this smaller firm-level sample size (Henseler and Sarstedt 2013). Additional information regarding the appropriateness of this method is outlined in the analysis. As illustration of the model is presented in Figure 1, the model is tested. Within this section the sample characteristics of the companies sampled for this research are outlined. Additionally, the development of the questions for the research instrument is explicated within the section.

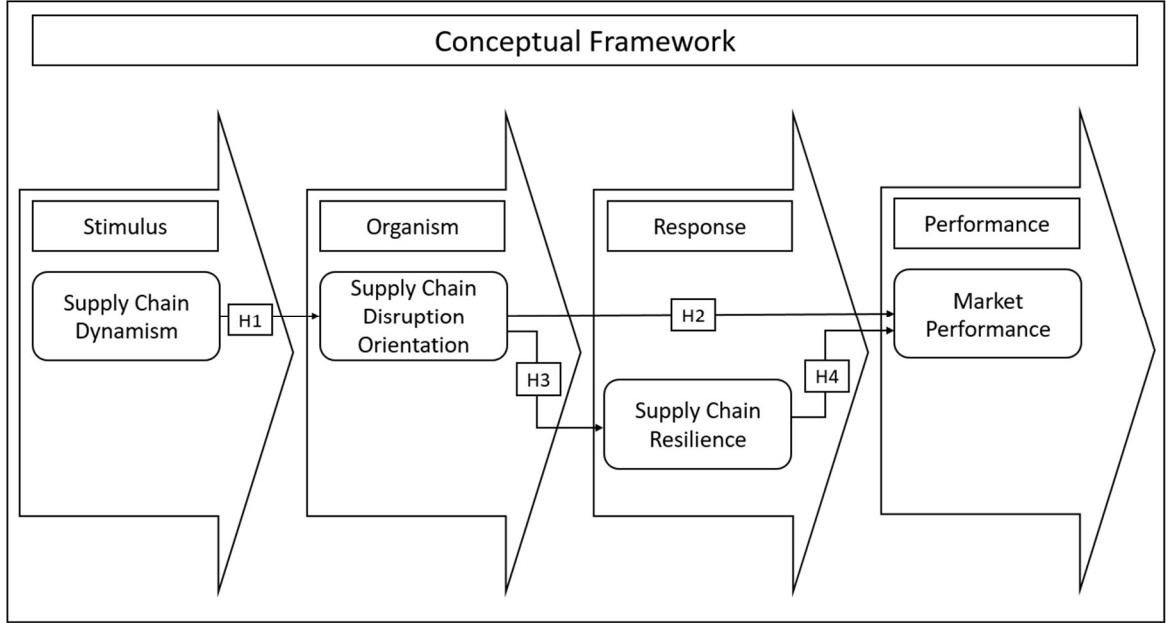

**Figure 1.** Conceptual Framework.

*3.1. Sample*

The sample was collected in April of 2021, an ideal period because firms would have experienced supply chain disruptions and had time to both respond and develop some degree of supply chain resilience. Furthermore, American firms were determined to be of interest because they would have endured supply chain disruptions. American firms also make up a segment of the world market that is frequently followed by both researchers and practitioners globally. One thousand American companies were sent an online survey by a research company that specializes in business-level data collection. As simple random sampling was employed, several types of bias were avoided. Of the 245 surveys returned, 227 responses were complete and accepted for analysis. Most companies (189 firms, 83%) had a turnover of less than USD 50 billion. Additionally, regarding the number of employees at the firm, almost half of companies (117 firms, 51.5%) are SMEs that had between 21 and 499 employees. Microfirms (69 firms, 30%) made up a larger portion than large firms (41 firms, 18%). Regarding age, most companies (78%) had been in operation for more than six years. Fewer companies were in operation for less than 6 years, indicating a more mature set of firms. Indeed, 59 firms had been in operation for more than 26 years making up a quarter of the respondents. Nearly all companies, whether service-based or otherwise, must deal with supply chain management. Unfortunately, the survey did not reveal more regarding the industry demographics; only a little more than a quarter of surveyed firms could be placed in an industry. The remaining 165 firms (72%) were not placed within the industries outlined by the survey. Please see Table 1 for a summary of the demographics.

**Table 1.** Demographics of the Sample.

| Number of Employees | | | | | |
|---|---|---|---|---|---|
| Interval | Less than 20 | 21–149 | 150–249 | 250–499 | 500+ | Total |
| Count (%) | 69 (30%) | 52 (24%) | 46 (20%) | 19 (8%) | 41(18%) | 227 (100%) |
| **Number of Years in Operation** | | | | | |
| Interval | 1 to 5 years | 6 to 10 | 11 to 25 | 26+ | | Total |
| Count (%) | 51 (22%) | 62 (28%) | 55 (24%) | 59 (26%) | | 227 (100%) |
| **Annual Sales** | | | | | |
| Interval | USD 5 mil. or less | USD 5–10 mil. | USD 10–20 mil. | USD 20–50 mil. | 50 mil.+ | Total |
| Count (%) | 90 (40%) | 26 (12%) | 36 (16%) | 37 (16%) | 38 (16%) | 227 (100%) |
| **Industry** | | | | | |
| Industry Type | Machinery, automobiles | Building materials | Chemical and petrochemical | Electronics and electrical | Others | Total |
| Count (%) | 13 (6%) | 15 (7%) | 14 (6%) | 20 (9%) | 165 (72%) | 227 (100%) |

*3.2. Research Instrument*

The research instrument is made up of both demographic questions and psychosomatic questions that measure the following constructs: supply chain dynamism, supply chain disruption orientation, supply chain resilience and market performance. The questions from each of constructs were taken from extant literature and modified to fit the context of this supply chain management research amid COVID-19. Additionally, the questions were set to a 5-point Likert scale because the respondents are generally familiar with it (Mandarić et al. 2022), and it is also the most frequently employed resolution for research instruments. The evolution and origin of each variable is explicated in the following paragraphs. Please see Table 2 regarding to the survey questions and variables with references.

**Table 2.** Operationalization of the Research Instrument.

| Variable | Operational Definition | Measurement Items | Prior Research |
|---|---|---|---|
| Supply Chain Dynamism | The degree to which supply chains are changing. | 〔SCD1〕At my company, new products account for most of total revenue.<br>〔SCD2〕At my company, products and services are changed frequently.<br>〔SCD3〕At my company, operations become outdated quickly.<br>〔SCD4〕At my company, unexpected and disruptive events happen frequently (e.g., shocks and disruptive technologies). | Zhou and Benton (2007) |
| Supply Chain Disruption Orientation | The degree to which an organization learns from and prepares for SC disruptions. | 〔DO1〕At my company, we are alert for possible supply chain disruptions at all times.<br>〔DO2〕At my company, we expect supply chain disruptions are always looming.<br>〔DO3〕At my company, we think about how supply chain disruptions could have been avoided.<br>〔DO4〕At my company, after a supply chain disruption has occurred, it is analyzed thoroughly. | Bode et al. (2011) |

**Table 2.** *Cont.*

| Variable | Operational Definition | Measurement Items | Prior Research |
|---|---|---|---|
| Market Performance | The degree to which this firm is able to perform well within the market. | (MP1) Comparing with our major competitor(s), our firm has higher/better customer loyalty. (MP2) Comparing with our major competitor(s), our firm has higher/better customer satisfaction. (MP3) Comparing with our major competitor(s), our firm has higher/better company image. (MP4) Comparing with our major competitor(s), our firm has higher/better growth in market penetration. (MP5) Comparing with our major competitor(s), our firm has higher/better growth in industry competitiveness. | Carey et al. (2011) |
| Supply Chain Resilience | The degree to which a firm maintains its supply chain operations even amid disruptions. | (SR1) Our firm's supply chain can quickly return to its original state after being disrupted. (SR2) Our firm's supply chain has the ability to maintain a desired level of connectedness among its members at the time of disruption. (SR3) Our firm's supply chain has the ability to maintain a desired level of control over structure and function at the time of disruption. (SR4) Our firm's supply chain has the knowledge to recover from disruptions and unexpected events. (SR5) Our firm's supply chain is well prepared to deal with the financial outcomes of supply chain disruptions. (SR6) Our firm's supply chain can move to a new, more desirable state after being disrupted. | Golgeci and Ponomarov (2013) |

### 3.3. Supply Chain Dynamism

Supply chain dynamism refers to the level of dynamism (change) within a focal firm's supply chain (Zhou and Benton 2007). Zhou and Benton (2007) outlined four key points regarding supply chain dynamism: (1) the degree to which new products contribute to revenue, (2) the degree to which products change (more frequent change indicates increased dynamism), (3) the degree to which operations change (increased change indicates increased dynamism, and (4) the degree to which supply chain disruptions occur (more frequent disruptions that last longer indicates greater dynamism). The combination of these factors was used by Yu et al. (2019) to measure supply chain dynamism while studying Chinese firms.

### 3.4. Supply Chain Disruption Orientation

Supply chain disruption orientation is a strategic orientation of the firm that suggests a company is both ready and able to manage disruptions within the supply chain (Bode et al. 2011). Bode et al. (2011) suggest that companies that experience frequent disruptions within the supply chain also exhibit feelings regarding those disruptions; they measure those feelings at several points, including (1) a sense that disruptions are inevitable, (2) the organization is alert and looking out for new disruptions, (3) the company has resolved disruptions, and (4) the company is able to learn from any disruption. Both Bode et al. (2011) and Yu et al. (2019) used this measure of supply chain disruption orientation.

### 3.5. Supply Chain Resilience

The measurement for supply chain resilience developed by Golgeci and Ponomarov (2013) was employed by multiple researchers over the years (Golgeci and Ponomarov 2014; Yu et al. 2019; Al-Hakimi et al. 2021). Golgeci and Ponomarov (2014) measured supply chain resilience at multiple points to measure the ability of the firm to maintain operations amid disruption and recover from the disruption afterwards. Additionally, they considered how the firm was able to return to normal, remain connected, maintain control, and improve after a disruption (Golgeci and Ponomarov 2014). In summary, supply chain resilience measures the ability of a firm to perform amid a supply chain disruption and return to normal or better operational performance afterward.

### 3.6. Market Performance

Market performance was adopted for this study because it remains a more neutral question for most managers compared to questions about financial performance. Objective financial performance questions usually elicit apprehension from management. The close link between market performance and financial performance suggests that either could be an excellent measure of financial performance. It is also the case that market performance also frequently impacts other areas of performance (Green et al. 2012). Market performance considers the ability of a firm to perform within the markets where it sells its products and services. Amid supply chain disruptions, it is anticipated that market performance can suffer; therefore, it is necessary to measure market performance. Although market performance is often used, it can be defined differently by various researchers. For this research study, a measurement adopted by Wong et al. (2020) and created by Kim (2009) was adopted. Accordingly, three measures of market performance were measured: customer loyalty, customer satisfaction, and corporate image (Kim 2009).

## 4. Analysis

Partial least squares structural equation modeling (PLS-SEM) has emerged as a valued analysis method for business research because of its ease of use and its ability to provide significant results at smaller sample sizes (Henseler and Sarstedt 2013). For this empirical study, PLS-SEM was utilized to measure the model and the impacts of the independent variable upon dependent variables. PLS-SEM is the most appropriate analysis method given the smaller sample size that would not be appropriate for covariance-based structural equation modelling using AMOS (Ringle et al. 2012; Barclay et al. 1995). According to Barclay et al. (1995), a sample size should be either ten times the largest number of items measuring a construct or ten times the largest number of pathways aiming at a construct; accordingly, a sample size for this model should be at least sixty respondents. With over 200 respondents, the minimum number of respondents was met. The following section includes the PLS-SEM analysis.

### 4.1. Outer-Model Assessment

While conducting SEM analysis, it is necessary to confirm the reliability and validity of the outer model (the questions representing the variables) before looking at the structural characteristics (the interrelationships between the variables); moreover, it is necessary to confirm that the items gauge the constructs that they were meant to measure (Hair et al. 2014). Two values validate reliability including both Cronbach's alpha and composite reliability; furthermore, both numbers are a quantification of internal consistency reliability (Hair et al. 2014; Nunnally and Bernstein 1994). Nunnally and Bernstein (1994) recommend that values for composite reliability be above 0.5 and values for Cronbach's alpha be above 0.6. Both cut-off values are met by the corresponding numbers; thus, reliability is established. Please see Table 3 for results of the outer model assessment.

**Table 3.** Outer Model Assessment.

| Variable | Factors | Standard Load | AVE (AVE > 0.5) | Construct Reliability (C.R > 0.7) | Cronbach's Alpha ($\alpha$ > 0.6) |
|---|---|---|---|---|---|
| Supply Chain Dynamism | SCD1<br>SCD2<br>SCD3<br>SCD4 | 0.754<br>0.827<br>0.719<br>0.710 | 0.569 | 0.840 | 0.749 |
| Supply Chain Disruption Orientation | SCDO1<br>SCDO2<br>SCDO3<br>SCDO4 | 0.808<br>0.757<br>0.874<br>0.808 | 0.661 | 0.886 | 0.828 |
| Market Performance | MP1<br>MP2<br>MP3<br>MP4<br>MP5 | 0.749<br>0.766<br>0.759<br>0.715<br>0.808 | 0.578 | 0.872 | 0.817 |
| Supply Chain Resilience | SCR1<br>SCR2<br>SCR3<br>SCR4<br>SCR5<br>SCR6 | 0.736<br>0.778<br>0.774<br>0.779<br>0.690<br>0.655 | 0.543 | 0.877 | 0.831 |

According to Hair et al. (2014), ensuring validity of the outer model requires two measures: (1) a measure of convergent validity, average variance extracted (AVE), and (2) a measure of discriminant validity, the Fornell and Larcker Criterion Test. Additionally, cross-loadings can be reviewed to confirm discriminant validity although Henseler et al. (2009) suggested that the Fornell and Larcker Criterion Test is a stricter test of discriminant validity. The Fornell and Larcker Criterion Test requires that the square root of the AVE numbers be larger than the latent variable correlation values (Fornell and Larcker 1981). Based on both cross loadings and the Fornell and Larcker Criterion Test, discriminant validity can be authenticated for the outer model. The results of these findings are presented in Table 4.

**Table 4.** Fornell–Larcker Criterion.

|  | MP | SCD | SCDO | SCR |
|---|---|---|---|---|
| MP | **0.760** | | | |
| SCD | 0.306 | **0.754** | | |
| SCDO | 0.454 | 0.557 | **0.813** | |
| SCR | 0.653 | 0.349 | 0.588 | **0.737** |

MP: market performance; SCD: supply chain dynamism; SCDO: supply chain disruption orientation; SCR: supply chain resilience.

### 4.2. Inner-Model Assessment

Once the outer model has been successfully assessed and confirmed, the researcher can move to assess the inner model (Hair et al. 2014). The inner model's assessment should involve testing the interrelationships of the variables to determine their impacts; moreover, the hypothesized pathways must be measured and tested for significance (Hair et al. 2014). Pathways that are not significant should be rejected while those that are significant should be accepted. PLS-SEM pathway analysis required two steps: (1) calculating the pathway coefficients and (2) calculating the significance scores for the pathways. The results of those two steps are available in Table 5, which presents pathway assessment and also in Figure 2 where the results of the study are presented. Bootstrapping to 2000 samples was used to assess the significance of the pathways. Hair et al. (2014) recommended rejecting

significance values above 0.05 before examining the pathway coefficients. Four hypotheses were tested with three accepted (H1, H3, and H4), and one was rejected, H2. The pathway coefficients suggest that strong impacts upon the interrelated variables. Hair et al. (2014) explained that the pathway coefficient is a percent of the total variance explained by the inner model. Supply chain dynamism impacted supply chain disruption orientation at (0.557). Supply chain disruption orientation increased supply chain resilience (0.588) while supply chain resilience led to improved market performance (0.590).

**Table 5.** Pathway Assessment.

| Hypotheses | Pathways | Pathway Coefficient | t-Stats | p-Value | Results |
|---|---|---|---|---|---|
| H1 | SC Dynamism → SC Disruption Orientation | 0.557 | 10.490 | 0.000 | **Accepted** |
| H2 | SC Disruption Orientation → Market Performance | 0.107 | 1.338 | 0.091 | Rejected |
| H3 | SC Disruption Orientation → SC Resilience | 0.588 | 11.843 | 0.000 | **Accepted** |
| H4 | SC Resilience → Market Performance | 0.590 | 6.922 | 0.000 | **Accepted** |

SC refers to supply chain.

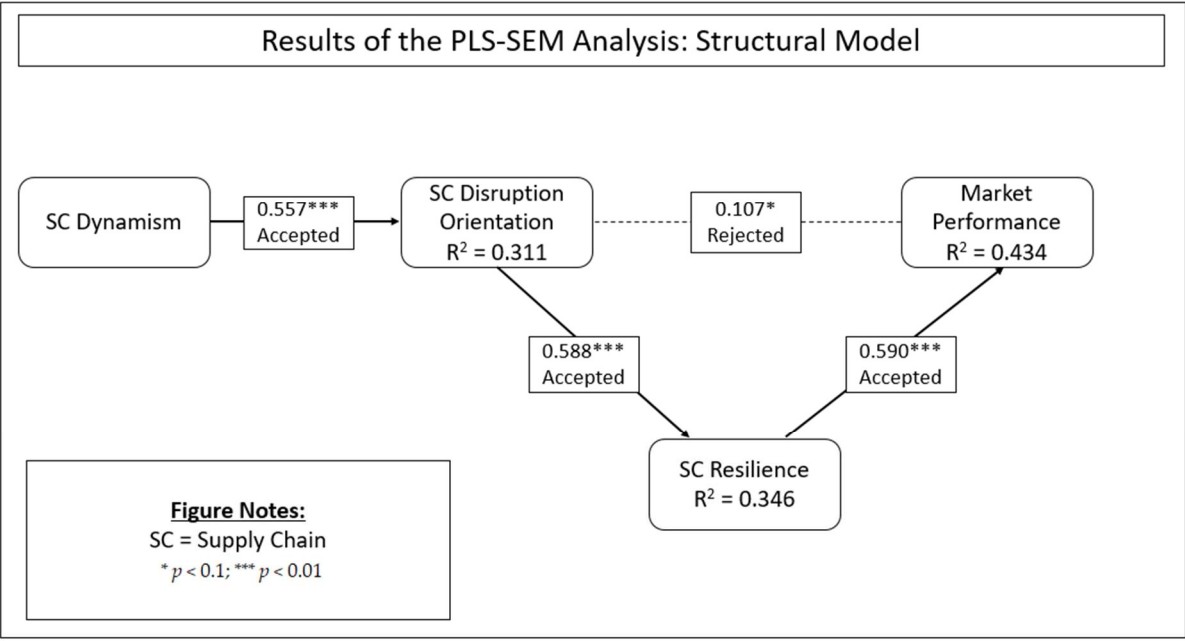

**Figure 2.** Results. Note: Figure 2 illustrates the pathway coefficients and the coefficients of determination for the variables of the research model. The asterisks indicate the *p*-value cut-off points: * is a *p*-value less than 0.1, and *** is a *p*-value less than 0.01.

The pathways indicate strong and meaningful relationships between the variables; however, it is important to test the strength of the results. The coefficient of determination $R^2$ is a percent of the variance explained by the model (Hair et al. 2014). The effects of the model can be measured as either small ($R^2 = 0.02$ up to 0.13), medium ($R^2 = 0.13$ up to 0.26), or large ($R^2 = 0.26$ and above) (Cohen 1988). According to the values for the coefficient of determination, the effects of the model are all large: supply chain disruption orientation (0.311), market performance (0.434), and supply chain resilience (0.346).

Cross-validated redundancy $Q^2$ indicates the predictive validity of the model; moreover, blindfolding is utilized to find $Q^2$ (Hair et al. 2014). Lately, any value above 0 indicates predictive validity (Hair et al. 2014). According to the values for $Q^2$, predictive validity is confirmed: supply chain disruption orientation (0.201), market performance (0.243), and supply chain resilience (0.185). Values for both Q2 and R2 can be reviewed in Table 6.

**Table 6.** Structural Model Assessment.

| Endogenous Variables | $R^2$ | $Q^2$ |
|---|---|---|
| Supply Chain Disruption Orientation | 0.311 | 0.201 |
| Market Performance | 0.434 | 0.243 |
| Supply Chain Resilience | 0.346 | 0.185 |

### 4.3. Assessment of Goodness-of-Fit

Finally, the global goodness-of-fit for the model should be evaluated. PLS-SEM lacks a standard measure of goodness-of-fit (GoF) that gauges the global model; however, two measures have emerged as proxies to capture a silhouette of GoF (Hair et al. 2014). Wetzels et al. (2009) suggest utilizing a process devised by Tenenhaus et al. (2005): the geometric mean of the average communality and the average $R^2$ for all endogenous constructs. According to Wetzels et al. (2009) such a quantification can establish GoF and indicate the degree of fit: small (0 to 0.10), medium (0.10 to 0.25), and large (0.36 and above). The GoF value for this model is 0.4264 (large). Additionally, another measure is standardized root mean square residual (SRMR) with a cut-off value of either liberally at less than 0.09 or strictly at less than 0.08 (Henseler and Sarstedt 2013; Hu and Bentler 1999). The value is 0.08 for this model; thus, another estimation of GoF is noted. Based on two commonly recommended values for GoF, we can confirm global fit. The results can be reviewed in Table 7 as the goodness-of-fit is shown.

**Table 7.** Goodness-of-Fit.

| Description | Value | Baseline Value | Reference |
|---|---|---|---|
| Goodness of Fit (GoF) | $\sqrt{\text{Cut} - \text{off of AVE X average of R\_square}} = \sqrt{0.5 \text{X } 0.364} = \mathbf{0.4264}$ | GoF *small* = 0.1<br>GoF *medium* = 0.25<br>GoF *large* = 0.36 | Wetzels et al. (2009) |
| | Standardized Root Mean Square Residual (SRMR) = 0.08 | Less than 0.08 | Hu and Bentler (1999) |

### 4.4. Mediation Effects

When conducting a SEM analysis, it is recommended that researchers consider the indirect effects of the model (Hair et al. 2014). Therefore, mediation tests were conducted. The most common mediation test is the Sobel test prescribed by Sobel (1982) (Nitzl et al. 2016; Cepeda-Carrion et al. 2018; Hair et al. 2014; Zhao et al. 2010). Hypotheses 5 (supply chain disruption orientation mediates the relationship between supply chain dynamism and supply chain resilience) and 6 (supply chain resilience mediates the relationship between supply chain disruption orientation and market performance) were proposed and tested by utilizing the Sobel test. Accordingly, mediation was validated with high significance scores for both tests. The results can be reviewed in Table 8:

**Table 8.** Mediation Effects.

| Mediating Pathways: | Mediation Effect (Z-Value) | *p*-Value |
|---|---|---|
| H5. Supply Chain Dynamism → **SC Disruption Orientation** → Supply Chain Resilience | 7.846 | 0.000 |
| H6. SC Disruption Orientation → **SC Resilience** → Market performance | 5.977 | 0.000 |

Mediating variables are in bold.

This research also measured the mediating effects of the constructs through the SOR framework. Both hypotheses 5 and 6 were accepted. Hypothesis 5 (Z = 7.846, *p* < 0.001) considered the interaction effects of disruption orientation in the relationship between supply chain dynamism and supply chain resilience. Additionally, hypothesis 6 deliberated



on the association between supply chain disruption orientation and market performance through the mediation of resilience in the supply chain (Z = 5.977, *p* < 0.001). These mediation results of the study suggest that having an orientation relative to supply chain disruption is important, and it can facilitate both the resilience of the supply chain and the market performance of organizations.

## 5. Discussion

Hypothesis 1, which measured the degree of change and dynamism on supply chain disruption orientation, was supported (β = 0.557, *p* < 0.001). These results are similar to research by Yu et al. (2019) who confirmed that turbulence in the external environment required firms to orientate their operations toward innovative initiatives that encouraged environmental learning and scanning to improve the overall performance of the supply chain. A debate still exists regarding the allocation of scarce organizational resources to alleviate the possible outcomes of supply chain risks (Al-Hakimi et al. 2021). However, this research finding suggests that organizations should be encouraged to embrace strategies that manage both the response and adaption to changes in the supply chain (Yu et al. 2019).

Interestingly, hypothesis 2, which considered the relationship between disruption orientation and market performance (β = 0.107, n.s), was not supported. Interestingly, the current research was able to confirm this relationship when mediated through resilience (see hypothesis 6). It, therefore, seems that merely committing to a process of disruption orientation does not necessarily improve the performance of an organization. In trying to understanding this study's outcome, we consider the research of Blackhurst et al. (2005) who suggested that the success on disruption orientation could be reliant on the current stock of an organization's resources or the ability of a firm to designate slack resources to manage supply chain disruptions. In this regard, firms are encouraged to find methods to amplify the functionality of their resources (Chowdhury and Quaddus 2017).

Hypothesis 3 reflected a positive (β = 0.588, *p* < 0.001) relationship between disruption orientation and resilience in U.S supply chains. This finding is maintained by Queiroz et al. (2021) who encouraged an organization to collectively pursue strategies that would result in approaches to build supply chain strength. Some authors (Bode et al. 2011; Yu et al. 2019; Zhou and Benton 2007) have suggested firm attitudes promoting information sharing among supply chain partners to promote resilience.

The affiliation between market performance and resilience (Hypothesis 4) was also recognized (β = 0.590, *p* < 0.001). This result further confirms the importance of supply chain resilience to build competitive advantages in the supply chain. The organizational response to bridging firm operations during supply chain disruptions remains an influential approach for creating positive performance outcomes for firms (Wong et al. 2020). These results encourage the notion that reliance can be regarded as a mechanism that motivates organizations to rapidly and accurately respond to changes in the business environment (Bode et al. 2011) as a means of establishing sustainable supply chain success (Craighead et al. 2020).

### 5.1. Practitioner Implications

This research is valuable for both marketing and management practitioners. An organization comprises both marketers and managers, especially with regard to supply chain management; however, few studies combine competent supply chain management with market performance. Supply chain studies frequently stop at supply chain performance or financial performance; nevertheless, amid abhorrent supply chain shortages, it has been well surmised—yet untested—that a company might gain market advantages by improving its supply chain resilience. This study positively links that assertion to evidence that supply chain resilience does improve market performance. Furthermore, mediation stresses the importance of supply chain resilience amid supply chain shortages (supply chain dynamism). Firms are competing amid a new, hyper-dynamism that requires additional agility. Strategically positioning an organization by developing a strategic orientation and supply chain

disruption orientation develops the necessary organizational preparedness for engendering supply chain resilience. Both pathway coefficients and mediation emphasize the crucial role of this strategic orientation for developing supply chain resilience amid dynamism; therefore, firms competing amid supply chain dynamism should develop an organizational culture that resembles supply chain disruption orientation.

Organizational culture is deeply rooted in the organization and is exhibited by employees and management. A strategic orientation is an aspect of that culture that is specifically focused on a strategic perspective. In the case of supply chain disruption orientation, the employees have dedicated themselves to a state of readiness and developed a degree of alertness to disruptions within the supply chain. As indicated by the results, employee readiness and attention to the supply chain facilitates supply chain resilience. Managers can build this strategic orientation by directing attention to supply chain readiness. Once more, as the results indicate, these firms have naturally developed this strategic orientation because of previous disruptions (supply chain dynamism). Firms that expect to undergo a period of supply chain dynamism could possibly develop both supply chain disruption orientation and supply chain resilience with training that builds the components of an organizational culture that is focused on the supply chain.

Finally, institutional implications should be approached with caution as the study heavily emphasizes organizational culture and firm behavior. Some institutional implications could be extended to investment in technology that facilitates supply chain transparency. More relevant to this study, investment in education or training that builds supply chain disruption orientation in other firms could build and lead to more competitive firms, especially amid supply chain dynamism.

### 5.2. Scholarly Implications

A number of academic implications can be concluded from the outcomes of the current research: (1) Increased dynamism within the supply chain requires added attention by scholars that can be further explored by the stimulus–organism–response model; (2) organizational culture plays a critical role in developing supply chain resilience and market performance; (3) this research theorizes that organizations compete within a hyper-dynamic environment that requires constant attention to environmental stimuli—such organizational behavior is more organic than organizational; thus, it more closely resembles the biological—researchers should evaluate organizational behavior and performance within this new hyper-dynamic paradigm. While the strategic outcomes of above-average returns incentivize firms to assess both the external and internal environment that an organization conducts operations within, the global context of value creation and organizational modeling has changed significantly since the introduction of COVID-19 (Katsaliaki et al. 2021). Moreover, since early 2020 and the spread of the pandemic (Laato et al. 2020), economic uncertainty has particularly increased, and previous assumptions regarding markets, mobility, and agglomeration procedures have reconditioned themselves (Rai 2020). Regarding supply chain management and marketing efforts, there has been an overt shift to adapting to a continuously changing external environment; therefore, it is necessary to have a theoretical model that fits with that reality, such as the stimulus–organism–response–performance model exemplified by this research. Firms are rushing to respond to the dynamism of this age; competitive firms respond by developing resilience to outperform competition. The SOR model fits better with the dynamism of companies competing today; furthermore, researchers that utilize this model have a valuable means of viewing the behavior of a firm within a hyper-dynamic environment that more resembles the organic world of creatures than that of competing organizations. Highlighting stimulus, organizational decision, response, and performance will characterize organizations well within the hyper-dynamism that has emerged.

Additionally, strategic orientations and the effects of organizational behavior given external stimuli can be further studied with both this conceptual framework and the strategic orientation, supply chain disruption orientation. Strategic orientations represent

the organizational mindset of firms well amidst the moment—going forward, supply chain disruption orientation will be a valuable means of understanding the strategic mind of a firm's organizational culture, especially regarding supply chain management.

To encourage more efficient supply chain activities such as disruption orientation and supply chain resilience, a more detailed understanding of supply chain dynamism has become necessary (Scholten et al. 2014; Shashi et al. 2020). Therefore, this research introduced supply chain dynamism as a stimulus that could influence the internal state of the organization. Due to the complexity with which modern supply chains operate, it is not surprising that an expected event occurs when interruptions occur (Lee et al. 2016). Dissimilarities exist between firms based on their ability to develop cognitive and affective processing of these dynamisms through the creation of strategic orientations (Matos and Krielow 2019). In the current research study, support was found for the first hypothesis, which stipulated that there is a positive relationship between supply chain dynamism and supply chain disruption orientation.

## 6. Conclusions

### 6.1. Contribution

This research study diversifies literature regarding supply chain resilience and market performance by utilizing the SOR model. According to the results of this study, several relationships are confirmed. Supply chain dynamism builds the strategic orientation supply chain disruption orientation, a strategic focus that emphasizes that an organization's culture is alert and ready to respond to supply chain disruptions. Furthermore, supply chain disruption orientation leads to supply chain resilience but not directly to market performance. Market performance is only bolstered when supply chain resilience is evident. Mediation results further highlight the magnitude of the effects between these variables. This study is the first to exhibit these relationships and conduct examinations with mediation effects. Additionally, this is conducted with a sample of U.S. firms, which would be especially valuable for U.S. firms hoping to improve supply chain resilience and market performance.

Academic implications are particularly strong regarding the use of the SOR model. Few firm-level studies have adopted this theory even if it has been in use for over four decades. It is especially relevant in this context as firms are struggling to respond to highly dynamic supply chains that closely resemble organic, life-like environments. Additional research in such highly dynamic environments should adopt this theory to frame organizational behavior.

### 6.2. Limitations and Future Research

While the study was conducted in a comprehensive manner, some limitations relative to this research study are noted. While consistent with the past literature (Bode et al. 2011; Yu et al. 2019), findings from the current research are limited with regards to sample demographics. Consequently, recreating the results with a larger sample size or through the introduction of an alternative sample location could present varying results. It is, therefore, understood that additional research be considered regarding a cross-country analysis or with the presentation of a greater sample size.

While this research has assessed supply chain operations and resilience under the COVID-19 pandemic, a comparison with past pandemics or disruptions could show interesting findings. For example, Rai (2020) noted that COVID-19 has acquainted organizations with 'very different' alternatives than previous crises regarding supply chain resilience, as the strategies of "deep freezing and reviving" organizational plans performed under preceding disasters are not possible under COVID-19 (Rai 2020).

Moreover, a possible direction for future research in supply chain resilience may be found in the analysis of supply chain partners in various tiers of the process or the introduction of digital technologies into the literature. Rai (2020) and Yu et al. (2019) mention that the incorporation of tier-2 suppliers into research could improve the understanding of supply-chain dependencies and requires a more impressive estimation of the outcomes of

risks on current solutions and practices (Shashi et al. 2020). A final limitation of the current research is related to the generalization of the supply chain in this study. The objective of this research was to ultimately assess market performance by using resilience. However, this overview could deliver deeper findings to supply chain literature in the future if the sample is specified to a certain industry. For instance, in healthcare, the demand for essentials such as ventilators or masks created overwhelming demands, which were more prevalent than in other industries. Finally, future research should acknowledge the importance of industry-specific indicators when conducting supply chain research.

**Author Contributions:** Conceptualization, A.R.S., M.K. and C.A.R.; methodology, M.K. and A.R.S.; software, M.K.; validation, A.R.S., M.K. and C.A.R.; resources, A.R.S., M.K. and C.A.R.; data curation, A.R.S., M.K. and C.A.R.; writing—original draft preparation, A.R.S., M.K. and C.A.R.; writing—review and editing, A.R.S., M.K. and C.A.R.; visualization, A.R.S., M.K. and C.A.R.; supervision, A.R.S., M.K. and C.A.R.; project administration, A.R.S., M.K. and C.A.R. All authors have read and agreed to the published version of the manuscript.

**Funding:** This research received no external funding.

**Institutional Review Board Statement:** Not applicable.

**Informed Consent Statement:** Not applicable.

**Data Availability Statement:** The data presented in this study are available upon request from the corresponding author.

**Conflicts of Interest:** The authors declare no conflict of interest.

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
