# Peer review of "Linking Supply Chain Disruption Orientation to Supply Chain Resilience and Market Performance with the Stimulus–Organism–Response Model"

_jrfm, doi:10.3390/jrfm15050227_

Round 1
Reviewer 1 Report
Please see below my comments and suggestions.
- The authors might want to add more descriptions about the surveyed and sampled firms.
- Is there a self-selection bias problem? The survey was sent in April 2021, and there could be many small firms that didn't survive after the pandemic and not responded to the survey.
- The study focuses on market performance, but how about other performance measures, such as financial performance.
Reviewer 2 Report
The article is interesting and generally, it deserves to be published with some revisions that are suggested below:
- Can you state clearly what the organizational culture plays a critical role in developing supply chain resilience and market performance amid supply chain dynamism is what, and what kind is the? (from your Results demonstrate)
- You need to describe what the strategic orientation is, and based on what, as practitioners have guidance? (in your conclusion)
- Would you please explain what the result after testing the 7 hypotheses are, and what values are, for the conclusion in Abstract and Conclusion?
Reviewer 3 Report
I have carefully read your manuscript " Linking supply chain disruption orientation to supply chain resilience and market performance with the stimulus-organism-response model " and I enjoyed it.
I think that the manuscript discreetly written and structured, addressing an interesting and important topic.
However, I believe that to accept the paper, the authors still need to focus on some changes or justify some choices underlying the research:
1) the authors should explain more about the adoption of the S-O-R framework and why they did not use other frameworks
2) the authors should better specify the choice of a sample of 1000 companies.
3) why is the sample collected in April 2021?
4) why were US companies chosen?
5) why the questions were set to a 5-point Likert scale?
6) the discussion of the results and the testing of research hypotheses should be broadened.
7) the contribution in terms of originality and innovativeness of the authors should be more explicit.
8) scientific, managerial and institutional (if there are) implications should be more explicit in the introduction
9) institutional implications should explicit (if there are) .
Good luck for your research!!!
Reviewer 4 Report
The subject of the article is very topical. The considerations are interesting, however, they should be supplemented with the objectives, research questions and hypotheses or thesis clearly marked in the text. This should be done in the first part of the discussion so that the reader knows what to expect. A statement of sub-hypotheses (as has been done by the authors) can be made in the text, but there is no coherent and well-defined aim of the research at the beginning of the text. The sub-theses are quite trivial (H1 and H2), H3 is interesting and worth exploring, but the authors devoted the least amount of space to it. There are too many sub-hypotheses - H4, as presented, does not really contribute anything.
Too much theory, I propose to limit the theory to the most important issues and refer to the literature instead of describing individual issues.
Lack of justification of the methodology, just indicating that it is based on literature is not enough. The structural modelling used in this area (in the literature) should also be referred to.
Conclusions rather trivial.
In the context of the use of structural equation modelling, there is no framework, which should be reflected later in the study and the final model.
Round 2
Reviewer 3 Report
Dear Authors
Thank you for taking my comments into account and improving the manuscript. The manuscript looks much better in its current form. I have no further comments and recommend the work for publication.
I wish you further success in your scientific work.